# One-Year Follow-Up Diagnostic Stability of Autism Spectrum Disorder Diagnosis in a Clinical Sample of Children and Toddlers

**DOI:** 10.3390/brainsci11010037

**Published:** 2021-01-01

**Authors:** Loredana Benedetto, Francesca Cucinotta, Roberta Maggio, Eva Germanò, Roberta De Raco, Ausilia Alquino, Caterina Impallomeni, Rosamaria Siracusano, Luigi Vetri, Michele Roccella, Massimo Ingrassia, Antonella Gagliano

**Affiliations:** 1Department of Clinical and Experimental Medicine, University of Messina, 98122 Messina, Italy; loredana.benedetto@unime.it (L.B.); robertaderaco@gmail.com (R.D.R.); massimo.ingrassia@unime.it (M.I.); 2Division of Child Neurology and Psychiatry, Department of the Adult and Developmental Age Human Pathology, University of Messina, 98122 Messina, Italy; cucinottafr@gmail.com (F.C.); maggioroberta19@gmail.com (R.M.); eva.germano@unime.it (E.G.); alquinoausilia@hotmail.it (A.A.); impallomenic@unime.it (C.I.); 3Division of Child Neurology and Psychiatry, Federico II University Hospital Naples, 80131 Napoli, Italy; rmsiracusano@yahoo.it; 4Department of Sciences for Health Promotion and Mother and Child Care “G. D’Alessandro”, University of Palermo, 90127 Palermo, Italy; 5Department of Psychology, Educational Science and Human Movement, University of Palermo, 90128 Palermo, Italy; michele.roccella@unipa.it; 6Child & Adolescent Neuropsychiatry Unit, Department of Biomedical Science, University of Cagliari, 09123 Cagliari, Italy; antonellagagliano.npi@gmail.com

**Keywords:** autism spectrum disorder, behavioral treatment, diagnostic stability, follow-up

## Abstract

Some studies show that the diagnosis of Autism Spectrum Disorder could be considered reliable and stable in children aged 18 to 24 months. Nevertheless, the diagnostic stability of early ASD diagnosis has not yet been fully demonstrated. This observational study examines the one-year diagnostic stability of autism spectrum disorder diagnosis in a clinical sample of 147 children diagnosed between 18 and 48 months of age. The ADOS-2 scores were used in order to stratify children in three levels of symptom severity: Autism (AD; comparison score 5–7), Autism Spectrum Disorder (ASD; comparison score 3–4), and Sub-Threshold Symptoms; (STS; comparison score 1–2). Results: Overall, the largest part of children and toddlers diagnosed with autism spectrum disorder between 18 and 48 months continued to show autistic symptoms at one-year follow-up evaluation. Nevertheless, a significant percentage of children with higher ADOS severity scores exhibited a reduction of symptom severity and, therefore, moved towards a milder severity class one year later. Conversely, the number of subjects of the STS group meaningfully increased. Therefore, at one-year follow-up a statistically significant (χ^2^(2) = 181.46, *p* < 0.0001) percentage of subjects (25.2% of the total) who had received a categorical diagnosis of Autistic Disorder or Autism Spectrum Disorder in baseline no longer met the criteria for a categorical diagnosis. Furthermore, children who no longer met the criteria for autism spectrum disorder continue to show delays in one or more neurodevelopmental areas, possibly related to the emergence of other neurodevelopmental/neuropsychiatric disorders. Overall, the comprehensive results of the study account for a high sensibility but a moderate stability of ASD early diagnosis.

## 1. Introduction

Autism Spectrum Disorder (ASD) is a neurodevelopmental disorder that describes a range of etiologies and clinical presentations [1]. The symptoms of ASD include early-onset difficulties in social communication and unusually restricted, repetitive behaviors and interests [2] with a great phenotypic heterogeneity that encompasses numerous comorbidities [3,4] and that reflects a complex multifactorial etiology [5]. The prevalence of ASD has dramatically increased in the last decades, reaching estimates of 1 subject in 59 as reported by the Centre for Disease, Control, and Prevention in the US [6]. This phenomenon is partly related to changes in diagnosis reporting practices [7] and partly due to the environmental factors potentially involved in the pathogenesis [8]. The early onset and lifelong social and adaptive impairments resulting from ASD remain a significant cause of morbidity in society [9].

Since ASD has been considered a severe and chronic disability for decades, several studies have demonstrated that early intervention can improve both short- and long-term outcomes of children with ASD [10,11,12,13,14,15,16]. Researchers agree that the early identification of ASD and its subsequent treatment can ultimately lead to better outcomes and quality of life [17,18,19,20,21,22] and to a significant decrease of long-term societal costs [23]. The most rapid gains in development and the greatest reduction in symptom severity appear to occur in the first two years of intervention, most notably in the first year [11,24,25]. Rate of learning in the early stages of intervention predicts later gains [26,27]. Thus, early identification of autism is viewed as vital to prognosis, since numerous studies have consistently shown that children who receive treatment at a younger age have better outcomes [28].

Current diagnosis of ASD can be reliably made as early as two years of age [29] based on a combination of the standardized diagnostic assessment data for infants and toddlers (e.g., Autism Diagnostic Observation Schedule-Toddler—ADOS-T; [30]) and expert clinical opinion [31]. However, some studies showed that the diagnosis could be considered reliable and stable in children aged 18 to 24 months [32,33,34,35,36]. The months surrounding the first birthday are considered a remarkable time for a toddler’s development [37]. At this age, toddlers learn to walk, speak their first words, and engage in a range of joint social attention behaviors, such as pointing and showing objects to others to share social attentional focus [38].

Although the awareness of early signs of autism may be increasing along with the prevalence of ASD and many children with ASD show developmental concerns in their educational and/or health records prior to age three, for example most balance-influencing parameters mature with age but not in ASD children [39], currently as few as 42% of children with autism at age eight have received their first comprehensive evaluation before their third birthdays, with median age of diagnosis at 40–53 months [6,40,41].

This diagnostic delay could be due, at least in part, to a series of unsolved questions regarding the early-age diagnostic stability, the age of clinical symptom onset and the overlap of early-age clinical symptoms between ASD and other disorders (e.g., language disorders or global developmental delay).

A meta-analysis of studies based on DSM-IV criteria had reported a significant stability (overall stability rate of 86.3%) of ASD diagnosis over time in children diagnosed before 36 months [42]. In addition, a systematic review of 10 studies conclude that diagnostic stability of ASD diagnosis is higher in toddlers diagnosed before age three [43]. New studies conducted after the DSM-5 modification of ASD inclusion criteria largely confirmed the stability of the early ASD diagnosis. For example, a recent study [44] aimed to prospectively follow 96 children, initially assessed for suspected ASD at an average age of 2.9 years and found that 76 met full criteria for ASD after two years of follow-up. Interestingly, the children who did not meet the criteria for ASD at T2 had symptoms of or met the criteria for other neurodevelopmental/neuropsychiatric disorders in combination with marked autistic traits.

However, clinicians and researchers have expressed concerns about the impact of the new DSM-5 criteria on the diagnosis of ASD in a population whose symptomatology may be emerging because the strict nature of the criteria is unsuitable for an early diagnosis model [45]. Moreover, enquiries about the permanence of ASD diagnosis have been emphasized by empirical reports of children who have lost the ASD diagnosis in middle or later childhood [46,47]. Concerns about the possibility to produce false positives are largely shared in the scientific community, and there are initial doubts about the stability of diagnosis for children identified before 36 months. Furthermore, both clinicians and researchers raised important questions about the youngest age at which a reliable diagnosis could be made, given the costs of early autism treatment [48].

Overall, the diagnostic stability of ASD diagnosis has not yet been fully demonstrated, even though much previous research supports the stability of ASD diagnosis even in very young children [33,34,35]. The present observational study aims to explore the one-year diagnostic stability of diagnosis in a clinical sample of toddlers and children diagnosed with ASD between 18 and 48 months of age. The purpose of the study is to evaluate if children would differ in diagnostic stability on the basis of the interaction between two variables: age of diagnosis and symptom severity.

## 2. Materials and Methods

### 2.1. Research Design and Procedure

We implemented a longitudinal prospective observational cohort study: first diagnostic evaluation (T0) and a follow-up evaluation (T1) at 12 months after the diagnosis. We recruited patients at their first neuropsychiatric consultation. This study was overseen by the Institutional Review Board of Intercompany Ethics Committee of the province of Messina, which approved the data collection and analysis (protocol number 56/13, 15 October 2013). Informed consent was obtained from both parents to data collection before the study enrollment, the patient names were removed from our spreadsheets to protect their identities. A multidisciplinary team (child neuropsychiatrists, psychologists, and speech/language therapists) with experience in ASD diagnosis conducted an extensive neuropsychiatric diagnostic assessment based on clinical observation and standardized measures. Parent clinical interviews about child’s developmental delays and/or problems (including play, social interaction, communication, and atypical behaviors) were carried out. A trained clinical psychologist conducted the evaluation of ASD severity symptoms and child’s developmental functioning with standardized measures (see below, Measures section). All the patients underwent the same assessment performed by the same professional team who collected standardized measures on child’s functioning and ASD severity at T0 and T1.

A complete diagnosis was assigned based on the clinical judgment of experienced clinicians in according to DSM-5 diagnostic criteria (APA 2013), throughout this longitudinal project, in order to maintain the same reliability. An extensive battery of clinical laboratory tests (EEG, auditory evoked potentials, genetic endocrine-metabolic, and immunological exams) was performed in order to exclude the association with known medical or genetic conditions or environmental factors, such as seizures and epilepsy, significant hearing and visual sensory deficits, traumatic brain injury or other significant genetic disorders (e.g., fragile X syndrome). At the conclusion of each evaluation, caregivers were provided with feedback regarding the assessment, which included the diagnosis and recommendations for intervention.

### 2.2. Participants

The inclusion criteria were patients aged between 18 and 48 months at their first neuropsychiatric consultation, availability of both Autism Diagnostic Observation Schedule (ADOS) baseline and follow-up scores. Children who presented other significant medical conditions (e.g., epilepsy, significant hearing and visual sensory deficits, traumatic brain injury, or other significant genetic disorders) were excluded.

We recruited 215 outpatients at their first neuropsychiatric consultation, aged between 18 and 48 months (M = 27.92, SD = 6.17), consecutively followed for ASD symptoms at the ambulatory of Child and Adolescent Psychiatry Unit of AOU Policlinico “G. Martino” of Messina, between June 2016 and January 2018. For 23 children (10.6%) a neurological, genetic, or metabolic disorder was found. Among the remaining 192 children, 25 (13%) dropped out for various reasons. Moreover, we excluded 20 children for whom the database was not completed.

Thus, the final research sample was of 147 children (117 males and 30 females). For a large percentage of children (38.1%), parents themselves directly requested a neuropsychiatric evaluation for concerns about their children’s social/communicative deficits, language delay and atypical/repetitive behaviors. A similar number of children (35.4%) was recruited by our unit after a primary care evaluation by the pediatrician and after a positive screening at the Modified Checklist for Autism in Toddlers (M-CHAT) [49] and another 23.1% was recruited after a specialist visit by an infant neuropsychiatrist and/or a neonatal neurologist. In 3.4% of the sample, parents were encouraged for accurate assessment by nursery educators/teachers who observed children’s impairments (i.e., social/communicative interactions, repetitive play, or restricted interests) in nursery/kindergarten settings. Children were Caucasians except for one child of African origins and all completed ADOS-2 assessment (see below) both at the first clinical diagnosis (T0) and at a follow-up assessment one year later (T1), (360 ±3 average days for participant).

To examine the stability of the clinical diagnosis related to the age of children, all participants were age-matched in months and divided into three classes: 18–24 months, 25–36 months, and 37–48 months

At baseline assessment (T0), all participants were grouped into three severity classes based on the ADOS-2 comparison score inferred from the appropriate comparison table provided in the ADOS-2 manual:Group 1 (*n* = 8) Sub-Threshold Symptoms (STS): children with isolated symptoms or very mild symptoms, not fully consistent with the criteria for the disorder (ADOS severity score between 1 and 2).Group 2 (*n* = 68) Autism Spectrum Disorder (ASD): children that still met the DSM-5 criteria for ASD but with mild symptom severity (ADOS severity score between 3 and 4).Group 3 (*n* = 71) Autistic Disorder (AD): children with the highest symptom severity that fully met the DSM-5 criteria for ASD (ADOS severity score between 5 and 7).

All children started the treatment within 3 weeks from the diagnosis. They received a usual intervention program specifically aimed to target ASD symptoms throughout the period between T0 and T1. All of them received the treatment as usual (TAU), available from local child neuropsychiatric services of their living area, for a mean of 6 (±1) h per week. TAU can be placed within a continuum ranging from highly structured behavioral approaches to approaches that follow the interests of the child in a naturalistic setting, mainly based on staff expertise rather than manualized treatment protocols. It also includes speech therapies for children and monthly parent coaching sessions. Overall, the type and the frequency of treatment were highly homogenous among the enrolled children.

### 2.3. Measures

#### Autism Severity

The Autism Diagnostic Observation Schedule—Second Edition (ADOS-2; [50]) is a semi-structured observation tool measuring ASD symptoms in children. A standard set of interactional examiner–child (or caregiver) activities was administered to assess social communication, play, repetitive, and restricted behaviors. It provides a measure of the severity of ASD symptoms, supporting clinical observation and decision about ASD diagnosis. Furthermore, it measures separately two main aspects: social affect (SA), restricted, repetitive behaviors (RRBs). It uses an algorithm composed by numeric score ranges from 0 to 2, with higher score indicating more severe deficits; the total score of SA and RRB can be transformed into a standardized severity score to compare ASD severity directly across modules.

In this study, the ADOS-2 scores were used in order to stratify children in three levels of symptom severity: Autism (AD; comparison score 5–7), Autism Spectrum Disorder (ASD; comparison score 3–4), and Risk for Autism (Sub-Threshold Symptoms; comparison score 1–2). The ADOS-2 consists of different modules with an activity program designed for children with different levels of language development. Noteworthy, ADOS was created for diagnostic purposes, and thus it was not specifically designed to facilitate longitudinal and cross-sectional data comparisons. In our study, we have used the ADOS as a diagnostic instrument, even though the total scores provided a common stand-in for a measure of autism severity.

In the present study Toddler Module (12–30 months) and Module 1 (31 months or older) were used for diagnosis at T0, whereas Module 1 and 2 (for verbally fluent children under three years) were administered at follow up (T1) according to the language level reached by the child. All modules were administered by trained examiners meeting the standard requirements for research reliability.

### 2.4. Statistical Approach

As first step, to test diagnostic stability we compared participant proportions within each severity class at T0 vs. T1 by chi-square test, using T0 proportions as expected values at T1.

Secondly, we distinguished four different patterns of stability: a stable positive diagnosis consisting of ASD or AD (true positives [TP]), a stable negative diagnosis (true negatives [TN]), and two unstable diagnostic groups including patients who met the ASD or AD criteria only at the first time point (false positives [FP]), and conversely patients who failed to meet the ASD or AD criteria at the first time point but received a diagnosis at the follow-up evaluation (false negatives [FN]). The McNemar’s test served to test these patterns of stability. Then, we computed sensitivity (as TP/(TP + FN) × 100), specificity (as TN/(TN + FP) × 100), positive predictive value (PPV as TP/(TP + FP) × 100), and negative predictive value (NPV as TN/(TN + FN) × 100). Differences in sensitivity and specificity between age groups were tested using the equiprobability chi-square test.

Moreover, we compared T0 and T1 ADOS-2 scores for each age class with the non-parametric Wilcoxon Signed Rank Test, because data did not approximate the characteristics of the normal distribution.

Statistical analyses were performed using the IBM Statistical Package for Social Science (SPSS), version 19.0.

## 3. Results

### 3.1. Severity Class Distribution at Baseline vs. Follow-Up (T0 vs. T1)

Table 1a,b shows the participants’ distribution within the three severity groups as function of the first neuropsychiatric consultation age at baseline (48.3% AD, 46.3% ASD; 5.4% STS) and at 1-year follow-up (32.7% AD, 36.7% ASD; 30.6% STS). The main result is the reduction of the percentage of subjects in the highest severity class from 48.3% to 32.7% (Figure 1). It does mean that 32.4% of the subjects with AD (23/71) moved towards a mild severity class one year later. In addition, the group with ASD had a slight percentage reduction. Conversely, the number of subjects of the STS group (children with isolated symptoms or very mild symptoms, not fully consistent with the criteria for the disorder) meaningfully (566.7%) increased from 5.4% to 30.6% of the total. Therefore, at one-year follow-up a statistically significant (χ^2^(2) = 181.46, *p* < 0.0001) percentage of subjects (25.2% of the total) who had received a categorical diagnosis of Autistic Disorder or Autism Spectrum Disorder in baseline no longer met the criteria for a categorical diagnosis.

The participants’ distribution within the three severity groups significantly changed from baseline to follow-up visit for each first-consultation-age group: 18–24 months, χ^2^(2) = 152.32, *p* < 0.0001; 25–36 months, χ^2^(2) = 66.09, *p* < 0.0001; 37–48 months, χ^2^(2) = 9.92, *p* < 0.007. In particular, in the first age class (18–24 months) we observed a significant enlargement of the STS group from 4.0% to 38.0% (χ^2^(1) = 17.25, *p* < 0.0001) and, conversely, a marked reduction from 66.0% to 34.0% of ASD group (χ^2^(1) = 10.14, *p* = 0.0015); in contrast to these two clusters, AD cluster undergoes minimal variations (χ^2^(1) = 0.05, *p* = 0.83).

### 3.2. Stability and Diagnostic Parameters (T0 vs. T1)

Table 2 shows the diagnostic trends from T0 to T1. Grouping AD and ASD participants into an “autistic class” (A) regardless of the age at the first consultation, we considered the four different patterns of stability: true positives (TP), true negatives (TN), false positives (FP), and, conversely, false negatives (FN) (see Table 3).

We estimated the same parameters along the three age groups and, from them, Sensitivity, Specificity, Positive Predicted Value (PPV), and Negative Predicted Value (NPV). These diagnostic parameters are summarized in Table 4 for participants taken as a whole, and in Table 5 for the age groups. The high sensitivity values contrast with the clearly lower specificity values; this reflects the small number of true negatives detected in the whole sample.

In Table 5 the 36–48 months stability rates were not reported because of the small sample size. Although sensitivity was almost the same between the two groups, there was a small increase in specificity from 18 months (5.26%) to 25 months (9.09%) and also in positive predictive values from 18 months (62.50%) to 25 months (75.30%). These proportion differences were not significant (all *p* > 0.05).

### 3.3. ADOS-2 Subscales at Baseline vs. Follow-Up (T0 vs. T1)

We compared follow-up outcomes after one year of follow up. From T0 to T1 all ADOS-2 total and sub-scale scores (Social Affect (SA) and Restricted and Repetitive Behavior (RRB)) decreased. Reductions were significant for SA scores of the two youngest age classes and for RRB scores in the oldest age class (see Table 6 and Table 7, and Figure 2). A significant reduction of symptoms was found in SA scores for the two youngest age classes and of RRB score symptoms in the oldest age class.

## 4. Discussion

The purpose of the present study was to examine the diagnostic stability of early autism diagnosis over twelve months. Overall, the largest part of children and toddlers diagnosed with autism spectrum disorder between 18 and 48 months continued to show autistic symptoms at one-year follow-up evaluation. Nevertheless, the participants’ distribution within the three severity groups significantly changed. In particular, the percentage of subjects in the highest severity class (AD) had a significant reduction. This means that a significant percentage of children with the ADOS highest severity score exhibited a reduction of symptom severity and, therefore, moved towards a mild severity class one year later. At the same time, the children with moderate symptom severity (ASD) had a slight percentage reduction. Conversely, the number of subjects of the STS group (children with isolated symptoms or very mild symptoms, not fully consistent with the criteria for the disorder) meaningfully increased from 5.4% to 30.6% of the total. Therefore, at one-year follow-up a statistically significant percentage of subjects who had received a categorical diagnosis of Autistic Disorder or Autism Spectrum Disorder in baseline no longer met the criteria for a categorical diagnosis.

This result is partially consistent with the previous literature evidence showing that early diagnosis of ASD is relatively stable and reliable, even when formulated between 18 and 36 months of life [35,36,37,38,40]. A more recent study [51] reports an overall ASD diagnostic stability of 0.84 (95% CI, 0.80–0.87) after 12 months, in a very large cohort of toddlers with a median age at the first evaluation of 17.6 months. The study also reports that the diagnostic stability of ASD within the youngest age band (12–13 months) was lowest at 0.50 (95% CI, 0.32–0.69) but increased to 0.79 by 14 months and 0.83 by 16 months, suggesting that an ASD diagnosis becomes stable starting at 14 months of age.

Inconsistent results across the studies could be related to different diagnostic tools and/or to the parameters used to evaluate the stability. However, the majority of studies underline that an accurate diagnosis in toddlers allows clinicians to start early treatments as a fundamental resource for achieving a better outcome in autism spectrum disorder. A very recent review [52] addressing the outcomes of young children with ASD who started early interventions at a range of ages, a strong evidence that “earlier is better” with regard to interventions for young children with ASD.

### 4.1. Diagnostic Stability in Function of Different Age Classes Compared to Different Severity Levels

Although previous research demonstrated stability of ASD diagnosis even in very young children, very few studies assessed the ASD diagnostic stability in relation to different age classes compared to different severity levels. Our longitudinal study provides a more detailed description of the outcome analyzing the differences between the three age classes and the three severity levels of symptoms.

In the youngest age class (18–24 months), a significant percentage of children lost the diagnosis of ASD, as it is possible to infer by the enlargement of the STS group and the reduction of ASD group. Conversely, the AD group maintained exactly the same amplitude. This could support the hypothesis that those who showed very severe autistic symptoms at a very young age maintained at the follow-up evaluation the same severity level of symptoms and, therefore, the diagnosis of Autism. The high degree of symptom stability in this severity class provides further support to the legitimacy of assigning the autism spectrum diagnosis to children aged between 18 and 24 months when they show a severe symptom level. Conversely, the percentage of children between 18 and 24 months showing milder symptoms (ASD diagnosis) has dropped by half at the follow-up evaluation. Thus, the validity of early autism diagnoses for the children with a lower symptom severity level (defined here as ADOS severity score between 3 and 4) seems to be weaker compared to those formulated for children showing clear autistic features with higher symptom severity ab initio. We argue that some behaviors could be abnormal because of the occurrence of different neurodevelopment disorders (e.g., ADHD, language disorder, ID) which are not easily distinguishable from autism in very young children. For instance, some authors suggest [53] that restricted and repetitive behaviors (RRBs) are common in children with Intellectual Disability (ID), leading to potential invalid autism diagnoses. Other authors [54] endorse the intellectual disability as a confounding factor that could potentially affect the validity of ASD diagnosis. The children of our sample losing the diagnosis of ASD at follow-up evaluation could have kept neurodevelopmental delay symptoms, as revealed by the enlargement of STS group (from 4.1% to 13.6%). These symptoms could be revealed by ADOS even if unreliable with a specific autistic phenotype. This is consistent with some authors’ description of autistic behaviors that may suggest a disease different from “classical” autism [55] and with most of the neurodevelopmental disorders sharing common features in cognitive, adaptive, social symptoms and stereotyped activities. According to some researchers’ hypothesis [56] the ASD diagnosis may be a “preliminary diagnosis” that will be changed as the child matures and more testing is done. A 2012 study, led by a government epidemiologist [57], found that 4% of children lost their diagnosis by age 8.2 and that almost all of them had at least another diagnosis, such as Attention-Deficit Hyperactivity Disorder (ADHD), developmental delay, or language delay. This kind of outcome could be more likely when the early ASD diagnosis is based on a milder presentation of symptoms. Therefore, below 24 months of age caution should be paid in formulating the ASD diagnosis in children presenting mild symptoms.

Another noteworthy point arises from the comparison between the outcomes of the children diagnosed between 18 and 24 months and the outcomes of children diagnosed later. In the second age class (25–36 months), both the STS and ASD groups increased, whereas the AD group significantly decreased. Actually, around one-third of subjects diagnosed between 25 and 48 months that met the full ASD criteria moved to a milder or subthreshold phenotype one year later. Therefore, the diagnostic stability of autism diagnosis seems to diminish with age increasing. This result may appear unexpected. However, many cultural campaigns focused on autism, and directed to pediatricians and general population, consistently reduced the average age at referral bringing it below 24 months. It appears to be plausible that the children of this group were probably later followed and diagnosed by a specialist. In particular, higher IQ scores and better verbal and communicative abilities would have led both parents and pediatricians to have fewer early concerns, even in presence of clear autistic symptoms. This hypothesis could explain the better outcomes and would be consistent with the large number of researches outlining that starting to speak at a younger age and having higher IQ facilitate the recovery [46,58,59]. At the same time, it must be stated, that ASD symptoms, usually clearly evident at two years of age, in one-third of children appear around 24 months, with a regression of the first acquired skills [18]. Children with this onset patterns do not appear to present different behavioral phenotypes from children with earlier onset of symptoms [60].

Regarding the oldest age class (37–48 months) of our sample since both the STS and ASD groups increased, whereas the AD group significantly decreased. Noteworthy, the sample of this age class is very small due to the low probability that the first ASD diagnosis is done at this age in our region. Therefore, the results of our study are not fully consistent with the previous reports supporting that children diagnosed with ASD at 30 months or younger were more likely to have a change in classification from ASD to non-ASD than children diagnosed with an ASD at 31 months or older [57]. However, the paucity of the oldest age class sample suggests considering this last evidence with caution.

### 4.2. Diagnostic Reliability in Relation to Different Age Classes

Our study also evaluated some parameters concerning the diagnostic reliability between different age classes. Consistently with previous literature evidence [48], our results show a good sensitivity of the early diagnosis. We found four different patterns in the three diagnostic classes (see Table 2): the first two patterns describe subjects with very clear diagnostic features (true positive or true negative). Conversely, the two other patterns describe children with unstable diagnosis: those who meet the ASD or AD criteria only at the first time point (false positives) or only at follow-up evaluation (false negatives) (see Table 3). Starting from these results and analyzing the sensitivity (the percentage of those diagnosed at the second time point who were identified at the earlier visit) and the specificity (the percentage of those without a diagnosis at the second time point who were correctly identified as STS at the first visit), we calculated a Positive Predicted Value. This value can be considered a more reliable parameter of the stability of the diagnosis since it describes the percentage of those identified with AD and ASD at the earlier visit who maintain the diagnosis at the second time point. Secularly, the Negative Predicted Value represents the percentage of those identified as STS at the first visit and failed to meet ASD criteria at the second time point (Table 4). Interestingly, the number of true negatives in the whole sample was very small. This could explain the high sensitivity and the relatively low specificity that we found. Finally, we found that the diagnostic sensitivity in relation to the age groups was very similar between the two youngest age classes, with only a slight advantage (not statistically significant) for the specificity and the positive predictive value for the children aged between 18 and 24 months. This result refers to age-dependent differences in the diagnostic stability of ASD/AD diagnosis with a favorite window for the identification between 18 and 24 months.

### 4.3. Follow-Up Outcomes Assessed by the ADOS-2 Subscale Scores at Baseline and Follow-Up

A second level of information provided by our study regards the follow-up outcomes assessed by the ADOS-2 subscale scores (Social Affect (SA) and Restricted and Repetitive Behaviors (RRB]) at baseline and follow-up. The focus on the symptom domains rather than on the overall symptom severity allows us to describe the diagnostic stability from a different perspective. In the two youngest age classes (18–24 and 25–36 months), the reduction of ASD total score was due exclusively to a reduction of SA scores. Conversely, Restricted and Repetitive Behaviors persisted at the same level. The opposite condition has been observed n the oldest age class (37–48 months), where the social affect domain does not further improve and only the RRB symptoms show a slight decrease.

Our results seem to be consistent with the conclusions of a recent review [61] aimed to examine the temporal stability of autism spectrum disorder and the longitudinal trajectories of autism core symptom severity. In fact, most of the included studies showed minor but statistically significant changes in ADOS total raw scores but no improvements in restricted and repetitive behaviors over time. At the same time minor improvements in social affect over time were described. These results could be related to the emphasis that the current early start treatment programs for ASD children have on the development of social competences, currently seen as the pivotal developmental domain. Most of the recent developed toddler interventions are delivered in naturalistic and interactive social contexts, and there is evidence that naturalistic interventions promote the social development in that they typically involve interactive exchanges between the child and an adult or typically developing peers [62,63].

By comparing the different diagnostic subgroups, we also identified the clinical characteristics that could help clinicians to outline the early diagnosis outcome. In particular, the earlier predicted factors for positive outcomes seem to be the symptom severity, fewer symptoms in RBBs domains, and stronger early social and communications skills.

## 5. Limitations and Future Directions

There are several limitations to the current study. First, the relatively small sample size of the enrolled subjects (mainly the 37–48 months group) limited the statistical power. Furthermore, the few amount of true negatives allowed a limited qualitative analysis. A second important limitation is that we cannot exclude some potential influences due to the qualitive differences of treatments received by children between T0 and T1. Moreover, our follow-up maximum age (48 months) limits our ability to assess children’s later outcomes. Future more prolonged studies, analyzing a larger group of children, with more homogeneous and manualized treatments, should give more information about the factors that could improve the diagnostic stability of ASD children.

## 6. Conclusions

The comprehensive results of the study account for a high sensibility but a moderate stability of ASD early diagnosis. One out of four children lost the diagnosis of ASD at one-year follow-up. Furthermore, children who no longer met the criteria for autism spectrum disorder continued to show delays in one or more neurodevelopmental areas, possibly related to the emergence of other neurodevelopmental/neuropsychiatric disorders.

Even if the stability of ASD diagnosis is moderately strong, clinicians ought to consider the early diagnosis and the access to an early treatment the strongest factor influencing the outcome.

## Figures and Tables

**Figure 1 brainsci-11-00037-f001:**
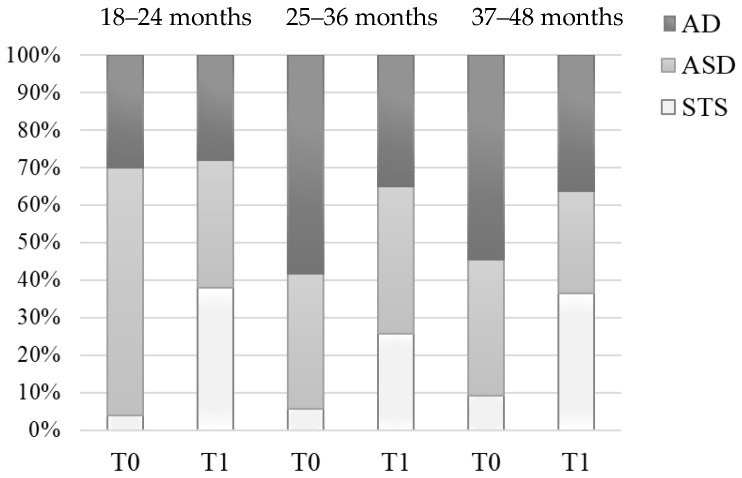
Participants’ distribution within the three severity groups as function of the age at baseline and follow-up visits.

**Figure 2 brainsci-11-00037-f002:**
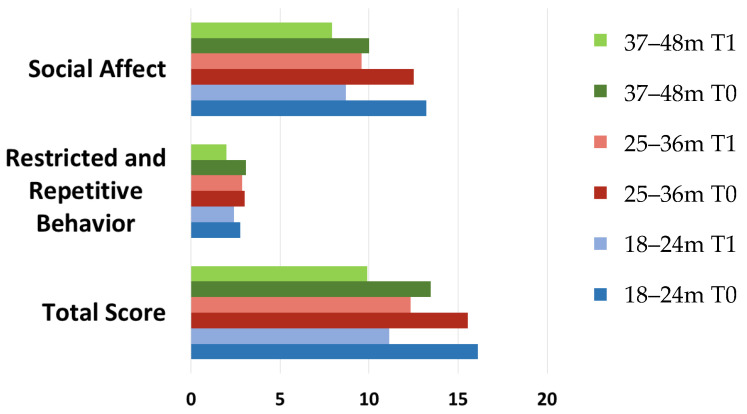
Age class and sub-scales (Social Affect, Restricted and Repetitive Behaviors, and Total) ADOS-2 scores at baseline and at 1-years follow-up. ADOS-2 = Autism Diagnostic Observation Scheme 0. = 1-year follow-up data collection.

**Table 1 brainsci-11-00037-t001:** Baseline (T_0_) and follow-up (T_1_) diagnosis categories by age classes.

a. (T_0_)			Age Classes			Total
			18–24 Months	25–36 Months	37–48 Months	
Baseline Diagnosis	**STS**	Counting	2	5	1	8
		T_0_-diagnosis %	25.0%	62.5%	12.5%	100.0%
		Age classes %	4.0%	5.8%	9.1%	5.4%
		Total %	1.4%	3.4%	0.7%	5.4%
	**ASD**	Counting	33	31	4	68
		T_0_-diagnosis %	48.5%	45.6%	5.9%	100.0%
		Age classes %	66.0%	36.0%	36.4%	46.3%
		Total %	22.4%	21.1%	2.7%	46.3%
	**AD**	Counting	15	50	6	71
		T_0_-diagnosis %	21.1%	70.4%	8.5%	100.0%
		Age classes %	30.0%	58.1%	54.5%	48.3%
		Total %	10.2%	34.0%	4.1%	48.3%
**b. (T_1_)**						
Follow up Diagnosis	**STS**	Counting	19	22	4	45
		T_1_-diagnosis %	42.2%	48.9%	8.9%	100.0%
		Age classes %	38.0%	25.6%	36.4%	30.6%
		Total %	12.9%	15.0%	2.7%	30.6%
	**ASD**	Counting	17	34	3	54
		T_1_-diagnosis %	31.5%	63.0%	5.6%	100.0%
		Age classes %	34.0%	39.5%	27.3%	36.7%
		Total %	11.6%	23.1%	2.0%	36.7%
	**AD**	Counting	14	30	4	48
		T_1_-diagnosis %	29.2%	62.5%	8.3%	100.0%
		Age classes %	28.0%	34.9%	36.4%	32.7%
		Total %	9.5%	20.4%	2.7%	32.7%
Total		Counting	50	86	11	147
		Age classes %	100.0%	100.0%	100.0%	100.0%
		Total %	34.0%	58.5%	7.5%	100.0%

Note: T_0_ = Time 0 (Baseline data collection), T_1_ = Time 1 (1-year follow-up data collection), AD = Autistic Disorder, ASD = Autism Spectrum Disorder, STS = Sub-Threshold Symptoms.

**Table 2 brainsci-11-00037-t002:** Trend over time of the three diagnostic classes.

	T_0_	T_1_		
		AD	ASD	STS
**AD**	71	32		
			28	
				11
**ASD**	68	15		
			23	
				30
**STS**	8	1		
			3	
				4
**Total**	147	48	54	45

Note: T_0_ = Time 0 (Baseline data collection), T_1_ = Time 1 (1-year follow-up data collection), AD = Autistic Disorder, ASD = Autism Spectrum Disorder, STS = Sub-Threshold Symptoms.

**Table 3 brainsci-11-00037-t003:** Outcome classification.

T_0_	T_1_	Total	Classification
A	A	98 (66.7%)	True positives
A	STS	41(27.9%)	False positives
STS	A	4 (2.7%)	False negatives
STS	STS	4 (2.7%)	True negatives
Total		147	

Note: T_0_ = Time 0 (Baseline data collection), T_1_ = Time 1 (1-year follow-up data collection), A = Autistic Disorder plus Autism Spectrum Disorder, STS = Sub-Threshold Symptoms.

**Table 4 brainsci-11-00037-t004:** Diagnostic reliability on the whole sample.

Diagnostic Parameters	
Sensitivity	96.08%
Specificity	8.89%
Positive Predictive Value	70.50%
Negative Predictive Value	50%

Note: Sensitivity is calculated as TP/(TP + FN) × 100; Specificity is calculated as TN/(TN + FP) × 100; PPV is calculated as TP/(TP + FP) × 100; NPV is calculated as TN/(TN + FN) × 100.

**Table 5 brainsci-11-00037-t005:** Stability and diagnostic classification parameters at different ages.

	True Positive (*n* = 98)	False Positive (*n* = 41)	False Negative (*n* = 4)	True Negative (*n* = 4)	Sensitivity	Specificity	Positive Predicted Value	Negative Predicted Value
18–24 months (*n* = 50)	30	18	1	1	96.77%	5.26%	62.50%	50%
25–35 months (*n* = 86)	61	20	3	2	95.31%	9.09%	75.30%	40%
36–48 months (*n* = 11)	7	3	0	1	--	--	--	--

Note: TP = True Positive, FP = False Positive, FN = False Negative, TN = True Negative; sensitivity is calculated as TP/(TP + FN) × 100; specificity is calculated as TN/(TN + FP) × 100; Positive Predicted Value (PPV) is calculated as TP/(TP + FP) × 100; Negative Predicted Value (NPV) is calculated as TN/(TN + FN) × 100.

**Table 6 brainsci-11-00037-t006:** T_0_ and T_1_ descriptive statistics of ADOS-2 sub-scales and total score as a function of the three age classes (18–24, 25–36, and 37–48 months at the first consultation).

ADOS-2 Scales	T_0_		T_1_	
	18–24 Months (*n* = 50)			
	M	SD	M	SD
Social Affect	13.22	3.65	8.68	5.20
Restricted and Repetitive Behavior	2.76	1.70	2.42	1.80
Total Score	16.10	4.51	11.14	6.16
	**25–36 Months** **(*n* = 86)**			
Social Affect	12.52	4.57	9.57	5.37
Restricted and Repetitive Behavior	3.02	1.75	2.87	2.03
Total Score	15.53	5.50	12.33	6.37
	**37–48 Months** **(*n* = 11)**			
Social Affect	10.00	4.49	7.91	4.11
Restricted and Repetitive Behavior	3.09	1.58	2.00	1.41
Total Score	13.45	5.47	9.91	5.32

Note: ADOS-2 = Autism Diagnostic Observation Schedule—Second Edition, T_0_ = Time 0 (Baseline data collection), T_1_ = Time 1 (1-year follow-up data collection).

**Table 7 brainsci-11-00037-t007:** Wilcoxon Signed Ranks Test applied to ADOS-2 SA and RRB sub-scores at T_0_ and T_1_ for each age class (18–24, 25–36, and 37–48 months at the first consultation).

18–24 Months	ADOS-2–SA T_1_–T_0_	ADOS-2–RRB T_1_–T_0_
*Z*	−4.384 ^a^	−0.783 ^a^
*p* (2-tailed)	< 0.001	0.646
**25–36 months**		
*Z*	−5.624 ^a^	−0.631 ^a^
*p* (2-tailed)	<0.001	0.528
**37–48 months**		
*Z*	−1.637 ^a^	−2.154 ^a^
*p* (2-tailed)	0.102	0.031

Note: ADOS-2 = Autism Diagnostic Observation Schedule—Second Edition, SA = Social Affect, RRB = Restricted and Repetitive Behavior, T_0_ = Time 0 (Baseline data collection), T_1_ = Time 1 (1-year follow-up data collection). ^a^ Based on positive ranks.

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
