# Peer review of "One-Year Follow-Up Diagnostic Stability of Autism Spectrum Disorder Diagnosis in a Clinical Sample of Children and Toddlers"

_brainsci, 2021, doi:10.3390/brainsci11010037_

Round 1

Reviewer 1 Report

Benedetto et al conducted an observational study of diagnostic stability of autism in a sample of children ages 18-48 months at diagnosis and one year after diagnosis. They find moderate changes in diagnostic stability that were rather stable with age, with most changes in children who lose their diagnosis at follow-up. This is an important topic to investigate, the methods are solid and the authors have conducted quite extensive investigations at diagnosis, including genetic and neurological testing to exclude association of diagnosis with a known disease. The analyses seem solid and the many tables and figures help the reader to understand the trends in the results. However, I have several main concerns, mainly about the introduction and discussion not presenting sufficient information to allow the reader to properly evaluate the novelty of the results and compare them to current knowledge. Furthermore, an additional concern is the lack of information about the treatment that children have received in the year between initial diagnosis and follow-up. Finally, several minor comments are to be corrected. Most of the paper is well-written and clear, yet every so often there is a sentence that seems to be out of place or missing grammatical sense. Please see my detailed report here below:

Major concerns:

1) Missing information in the introduction and/or discussion: As a study that tests diagnostic stability, I would expect a lot more coverage of prior studies that covered similar content, including their findings and how the current study complements them. In particular, the introduction details only one (!) study that tested diagnostic stability in autism, under DSM-IV. Next, they present an argument that the transition to DSM-5 raises concerns about diagnostic stability, but no other study is cited, If this is the first study to test this - the authors should emphasize this as a strength and importance of the paper (same goes for the division to age groups, etc). Otherwise, they should mention all relevant studies, to allow the reader to compare results and understand how different the current results are from what is known in the field. A good example I found in the discussion - row 354. That was helpful and I'd love to see more studies covered so nicely.

Of note, the paper mentioned in the introduction (row 72) uses a "stability measure" that I would love to see calculated in the current study as well. It is helpful to compare findings when they are reflected by the same measures.

2)A related concern - the authors present quite a weak specificity of diagnosis: less than 10% of participants who were diagnosed as STS at T1 were correctly identified as such at T0 (Table 4). In other words, 41 of 147 participants were False Negative (Table3). To me, and I am not a clinician, that sounds astonishing. Yet in the abstract and discussion, the authors do not present this as particularly concerning. Without comparison to the literature, I have no tools to judge. 

3)A major concern is the treatment participants received during the year in between diagnoses. There is no information on this issue, except being mentioned in the limitations, yet it can be critical to the point of the study: could it be that children who received one type of treatment were more likely to change their initial diagnosis? To me it sounds that treatment type should be considered as a potential confounder of the effect. That is, it should be controlled. If the authors do not have this information, they should explicitly acknowledge that and I would advise them to make a point about why would that not be a critical caveat. What I mean is that it could be that treatment type should not be a concern, that the diagnosis should remain stable, that other studies have tested and found that we expect a certain stability measure regardless of the treatment. In other words, there could be a way to rationalize this, but it must be clearly stated.

4)Another point that was unclear to me - and this is a genuine question to the authors - both in the abstract and in the conclusion the following sentence appears: "...children who no longer met the criteria for autism spectrum disorder continue to show delays in one or more neurodevelopmental areas". Is this a finding? If so, I saw no support for it in the results section. I would want to ask the authors to kindly explain this. 

5)References: while this is probably an honest mistake, references in the text were given as numbers, whereas in the ref list, they were ordered differently (not by numbers and not alphabetically either) so that matching between in-text citation to its reference was impossible.

Moderate concerns:

6)Line 219-220: "Overall, the trend of the 18-24 months group appears clearly different from the other two first-consultation-age groups." --> this is not a clear conclusion from the analyses presented prior to this sentence. Further analyses are needed or the sentence should be more clearly phrased.

7)line 319: "This result is partially consistent with the previous literature evidence showing that early diagnosis of ASD is relatively stable and reliable, even when formulated between 18 and 36 months of life [35–39], allowing the clinicians to start early treatments as a fundamental resource for achieving a better outcome in autism spectrum disorder." --> while this directly relates to points 1 and 2, I had to bring this up here too. To me, it is unclear how these results are in partial consistency with the above. The authors must support such a claim with evidence from other papers, for example: perhaps there is a natural drift between groups that has been demonstrated earlier? Without supporting evidence, this statement seems like they are trying to forcefully fit their results with the conclusion they wish to obtain.

Minor concerns:

8)General structure and language: a)lines 62-66: I suggest breaking into two sentences; b)lines 66 and 67 could be the same paragraph; same for rows 184-185; c)line 74: "...a systematic review of 10 studies conclude that diagnostic stability of ASD diagnosis is higher in toddlers diagnosed before age 3 [42]". I suggest adding "than XXX" (I would assume "than in a later age", but it doesn't make sense to me). d)row 199: "It does mean that" --> "That means that.."; row 205: "in baseline" --> "at baseline"; 

9)IRB approval and consent should be stated more clearly (row 98).

10)row 40: latest CDC report estimates ASD prevalence as 1 in 54 in the US (https://www.cdc.gov/media/releases/2020/p0326-autism-prevalence-rises.html). Since links to references were broken (see point 5), I couldn't check the ref.

11)row 133 - M-CHAT wasn't earlier defined.

12)row 159- "and play" may be redundant.

13)Table 1: formatting: "b. T1" is no aligned with "a.T0". Also, I would advise adding thin horizontal lines to help better reading the different severities classes.

14)Table 3: I was missing % of each category.

15)lines 243-4: "the high sensitivity values contrast with the clearly lower specificity values that may reflect the small number of true negatives detected in the whole sample." --> why "may"? This needs to be stated more clearly.

16)I would like to see information about the exact times of follow-up per participants, if possible. Something like the average days (and standard deviations) from T0 to T1.

17)Table 5: mention N per row, please.

18)Figures: quality is low, and it seemed to me that error bars were missing, so this is critical to add. In addition, and this is a suggestion only, to me it would have been easier to see all three age-groups on the same bar-graph. For your consideration.

19)section 3.4: the first paragraph isn't clear. Please provide more context to the reader about which four classes, what is this analysis for, what are the variables measured. This sentence was particularly unclear to me: "however, in the 25-36 months class of age, we found three parameters (Griffiths Languages, Performance and Developmental Quotient) that differed from the others, although not significantly." --> please rephrase, helo the reader understand and be reminded of these variables.

20)lines 326-328: unclear, please rephrase.

21)line 345: remove "the" at the end; line 352: "and" missing between [52] and "with"? If not, I didn't understand the sentence, please rephrase.

22)line 389" "the whole of our results 388 shows a good sensitivity of the early diagnosis with a high sensitivity." --> please correct/rephrase

23)lines 390-1: "We found four different patterns in the three diagnostic classes (see Table 2): patients immediately diagnosed with or without ASD or AD having a higher level of stability." --> isn't it circular to say that TP and TN cases had higher stability? If I misunderstood, please rephrase or write back to clarify.

24)line 396: should be "without a diagnosis" instead of "diagnosed" to define specificity.

Author Response

09-Dec-2020

Dear Editor and Dear Reviewers,

I would like to thank the editor and reviewers for your valued comments and suggestions to the article. As you requested, we made all the necessary changes in our manuscript to address the reviewers’ concerns and we detailed below how the points raised by the referees have been accommodated. We highlighted the changes within the document by using the track changes mode in MS Word. From the changes made in the revised manuscript and responses provided below, I hope you are convinced that we have adequately addressed the reviewer’s concerns and made the paper better. If there are any further questions, please feel free to let me know.

Reviewer 1

Benedetto et al conducted an observational study of diagnostic stability of autism in a sample of children ages 18-48 months at diagnosis and one year after diagnosis. They find moderate changes in diagnostic stability that were rather stable with age, with most changes in children who lose their diagnosis at follow-up. This is an important topic to investigate, the methods are solid and the authors have conducted quite extensive investigations at diagnosis, including genetic and neurological testing to exclude association of diagnosis with a known disease. The analyses seem solid and the many tables and figures help the reader to understand the trends in the results. However, I have several main concerns, mainly about the introduction and discussion not presenting sufficient information to allow the reader to properly evaluate the novelty of the results and compare them to current knowledge. Furthermore, an additional concern is the lack of information about the treatment that children have received in the year between initial diagnosis and follow-up. Finally, several minor comments are to be corrected. Most of the paper is well-written and clear, yet every so often there is a sentence that seems to be out of place or missing grammatical sense. Please see my detailed report here below:

Major concerns:

1) Missing information in the introduction and/or discussion: As a study that tests diagnostic stability, I would expect a lot more coverage of prior studies that covered similar content, including their findings and how the current study complements them. In particular, the introduction details only one (!) study that tested diagnostic stability in autism, under DSM-IV. Next, they present an argument that the transition to DSM-5 raises concerns about diagnostic stability, but no other study is cited, If this is the first study to test this - the authors should emphasize this as a strength and importance of the paper (same goes for the division to age groups, etc). Otherwise, they should mention all relevant studies, to allow the reader to compare results and understand how different the current results are from what is known in the field. A good example I found in the discussion - row 354. That was helpful and I'd love to see more studies covered so nicely. Of note, the paper mentioned in the introduction (row 72) uses a "stability measure" that I would love to see calculated in the current study as well. It is helpful to compare findings when they are reflected by the same measures.

Answer: We added a reference of a more recent study (Kantzer AK, Fernell E, Westerlund J, Hagberg B, Gillberg C, Miniscalco C. Young children who screen positive for autism: Stability, change and "comorbidity" over two years. Res Dev Disabil. 2018 Jan;72:297-307. doi: 10.1016/j.ridd.2016.10.004. Epub 2016 Nov 3. PMID: 27818061) having a general structure and a purpose comparable with the previous one.

2)A related concern - the authors present quite a weak specificity of diagnosis: less than 10% of participants who were diagnosed as STS at T1 were correctly identified as such at T0 (Table 4). In other words, 41 of 147 participants were False Negative (Table3). To me, and I am not a clinician, that sounds astonishing. Yet in the abstract and discussion, the authors do not present this as particularly concerning. Without comparison to the literature, I have no tools to judge. 

Answer: We agree. This evidence could seem astonishing, but modifications in developmental profiles and symptoms severity are common in children with suspected autism: for example, Kantzer et al. (2018) found that 40% of children (under 3 years) with autistic traits at T1, but not full ASD, met the criteria for ASD at T2 (two years later).

3) A major concern is the treatment participants received during the year in between diagnoses. There is no information on this issue, except being mentioned in the limitations, yet it can be critical to the point of the study: could it be that children who received one type of treatment were more likely to change their initial diagnosis? To me it sounds that treatment type should be considered as a potential confounder of the effect. That is, it should be controlled. If the authors do not have this information, they should explicitly acknowledge that and I would advise them to make a point about why would that not be a critical caveat. What I mean is that it could be that treatment type should not be a concern, that the diagnosis should remain stable, that other studies have tested and found that we expect a certain stability measure regardless of the treatment. In other words, there could be a way to rationalize this, but it must be clearly stated.

Answer: thank you for this very significant suggestion; we enlarged the explanation of the type of treatment and stated that it was homogeneous for all the enrolled children (see lines 166-174)

4)Another point that was unclear to me - and this is a genuine question to the authors - both in the abstract and in the conclusion the following sentence appears: "...children who no longer met the criteria for autism spectrum disorder continue to show delays in one or more neurodevelopmental areas". Is this a finding? If so, I saw no support for it in the results section. I would want to ask the authors to kindly explain this. 

Answer: We agree that the expression “delays in one or more neurodevelopmental areas” sounds ambiguous and unclear. It would try to describe the presence of concerns and symptoms which are not strictly related to the autism phenotype but linked to a different kind of neurodevelopmental disorders. Since neurodevelopmental disorders are a spectrum of highly overlapped conditions, defining discrete diagnostic categories sometimes results very hard.  Nevertheless, we tried to better clarify what “delays in one or more neurodevelopmental areas” means, both in the abstract and in the text.

5)References: while this is probably an honest mistake, references in the text were given as numbers, whereas in the ref list, they were ordered differently (not by numbers and not alphabetically either) so that matching between in-text citation to its reference was impossible.

Answer: thanks for the suggestion. Now the reference list is correctly numbered.

Moderate concerns:

6)Line 219-220: "Overall, the trend of the 18-24 months group appears clearly different from the other two first-consultation-age groups." --> this is not a clear conclusion from the analyses presented prior to this sentence. Further analyses are needed or the sentence should be more clearly phrased.

Answer: Yes, it is an inappropriate comment. We deleted the sentence.

7)line 319: "This result is partially consistent with the previous literature evidence showing that early diagnosis of ASD is relatively stable and reliable, even when formulated between 18 and 36 months of life [35–39], allowing the clinicians to start early treatments as a fundamental resource for achieving a better outcome in autism spectrum disorder." --> while this directly relates to points 1 and 2, I had to bring this up here too. To me, it is unclear how these results are in partial consistency with the above. The authors must support such a claim with evidence from other papers, for example: perhaps there is a natural drift between groups that has been demonstrated earlier? Without supporting evidence, this statement seems like they are trying to forcefully fit their results with the conclusion they wish to obtain.

Answer: Thank you for this remark. The sentence sounded very raw. We enlarged the comment as follow:

“Inconsistent results across the studies could be related to different diagnostic tools and/or to the parameters used to evaluate the stability. However, the majority of studies underline that an accurate diagnosis in toddlers allows clinicians to start early treatments as a fundamental resource for achieving a better outcome in autism spectrum disorder. A very recent review (Towle et al, 2020), addressing the outcomes of young children with ASD who started early interventions at a range of ages, shows a strong evidence that “earlier is better” with regard to interventions for young children with ASD.”

Minor concerns:

8) General structure and language: a)lines 62-66: I suggest breaking into two sentences; b)lines 66 and 67 could be the same paragraph; same for rows 184-185; c)line 74: "...a systematic review of 10 studies conclude that diagnostic stability of ASD diagnosis is higher in toddlers diagnosed before age 3 [42]". I suggest adding "than XXX" (I would assume "than in a later age", but it doesn't make sense to me). d)row 199: "It does mean that" --> "That means that.."; row 205: "in baseline" --> "at baseline"; 

Answer: done

9)IRB approval and consent should be stated more clearly (row 98).

Answer: thanks for the suggestion, we changed the text as you can read below:

This study was overseen by the Institutional Review Board of Intercompany Ethics Committee of the province of Messina, that approved to collect and analyze data. Informed consent was obtained from both parents to data collection before the study enrolment, the patient names were removed from our spreadsheets to protect their identities.

10)row 40: latest CDC report estimates ASD prevalence as 1 in 54 in the US (https://www.cdc.gov/media/releases/2020/p0326-autism-prevalence-rises.html). Since links to references were broken (see point 5), I couldn't check the ref.

Answer: done

11)row 133 - M-CHAT wasn't earlier defined.

Answer: done

12)row 159- "and play" may be redundant.

Answer: done

13)Table 1: formatting: "b. T1" is no aligned with "a.T0". Also, I would advise adding thin horizontal lines to help better reading the different severities classes.

Answer: done

14)Table 3: I was missing % of each category.

Answer: done

15)lines 243-4: "the high sensitivity values contrast with the clearly lower specificity values that may reflect the small number of true negatives detected in the whole sample." --> why "may"? This needs to be stated more clearly.

Answer: We changed the sentence as follow: “The high sensitivity values contrast with the clearly lower specificity values; this reflects the small number of true negatives detected in the whole sample”

16)I would like to see information about the exact times of follow-up per participants, if possible. Something like the average days (and standard deviations) from T0 to T1.

Answer: We stated 360 ±3 average days per participant (line 148).

17)Table 5: mention N per row, please.

Answer: we added N per row and columns (table is considerably improved).

18)Figures: quality is low, and it seemed to me that error bars were missing, so this is critical to add. In addition, and this is a suggestion only, to me it would have been easier to see all three age-groups on the same bar-graph. For your consideration.

Answer: We put all the three age groups on the same bar graph and we improved the quality of the figure according to the editor’s suggestion.

19)section 3.4: the first paragraph isn't clear. Please provide more context to the reader about which four classes, what is this analysis for, what are the variables measured. This sentence was particularly unclear to me: "however, in the 25-36 months class of age, we found three parameters (Griffiths Languages, Performance and Developmental Quotient) that differed from the others, although not significantly." --> please rephrase, helo the reader understand and be reminded of these variables.

Answer: We are very sorry. This paragraph belongs to a previous version of the manuscript that encompassed the Griffiths data of the described sample. For several reasons, we decided to eliminate this data, but we forgot to delete the related paragraph.   

20)lines 326-328: unclear, please rephrase.

Answer: We tried to reorganize the sentence.

21)line 345: remove "the" at the end; line 352: "and" missing between [52] and "with"? If not, I didn't understand the sentence, please rephrase.

Answer: done

22)line 389" "the whole of our results 388 shows a good sensitivity of the early diagnosis with a high sensitivity." --> please correct/rephrase

Answer: We changed the sentence as follows: Consistently with previous literature evidence [46], the whole of our results shows a good sensitivity of the early diagnosis.

23)lines 390-1: "We found four different patterns in the three diagnostic classes (see Table 2): patients immediately diagnosed with or without ASD or AD having a higher level of stability." --> isn't it circular to say that TP and TN cases had higher stability? If I misunderstood, please rephrase or write back to clarify.

Answer: Yes, thank you very much. We changed the sentence as follows: “We found four different patterns in the three diagnostic classes (see Table 2): the first two patterns describe subjects with very clear diagnostic features (true positive or true negative). Conversely, the two other patterns describe children with an unstable diagnosis: those who meet the ASD or AD criteria only at the first time point (false positives) or only at follow-up evaluation (false negatives)

24)line 396: should be "without a diagnosis" instead of "diagnosed" to define specificity.

Answer: Yes, of course. We modified the sentence.

Reviewer 2 Report

Dear Authors,

Thank you for the opportunity to revise your manuscript.

The study is interesting and well written. Authors should follow the STROBE in order to organize their manuscript.  Vandenbroucke JP, von Elm E, Altman DG, Gøtzsche PC, Mulrow CD, Pocock SJ, Poole C, Schlesselman JJ, Egger M; STROBE initiative. Strengthening the Reporting of Observational Studies in Epidemiology (STROBE): explanation and elaboration. Ann Intern Med. 2007 Oct 16;147(8):W163-94. doi: 10.7326/0003-4819-147-8-200710160-00010-w1. PMID: 17938389. There is a mixture of methods and results, as well as in the discussion section there is a repetition of the results.

Please find below my comments:

Introduction

When introducing the clinical description of Autism, the Authors may find interesting and cite the following studies addressing gait behaviour and postural capacity in Autism Spectrum Disorder:  

Valagussa G, Balatti V, Trentin L, Piscitelli D, Yamagata M, Grossi E. Relationship between tip-toe behavior and soleus - gastrocnemius muscle lengths in individuals with autism spectrum disorders. J Orthop. 2020 Aug 20;21:444-448. doi: 10.1016/j.jor.2020.08.013. PMID: 32982098; PMCID: PMC7493131.

Perin C, Valagussa G, Mazzucchelli M, Gariboldi V, Cerri CG, Meroni R, Grossi E, Cornaggia CM, Menant J, Piscitelli D. Physiological Profile Assessment of Posture in Children and Adolescents with Autism Spectrum Disorder and Typically Developing Peers. Brain Sci. 2020 Sep 27;10(10):681. doi: 10.3390/brainsci10100681. PMID: 32992546; PMCID: PMC7601261.

Lines 84-92: Please state clearly the Objectives and the Hypotheses. The paragraph is hard to follow. Moreover, in the Materials and Methods, Statistical Approach, refer the analyses to the Objectives. This will increase the readability of the manuscript.

Line 95: “two-survey design” is misleading. This is a prospective observational cohort study

Line 112: How genetic conditions were ruled out?

Line 188: How sensibility, specificity, positive and negative predictive values were evaluated? Was used a ROC?  In the Results section should not be described how the analysis was performed. All the test e.g., Wilcoxon Signed Ranks Test, chi-square test should be presented here. 

Authors should report the behavioral treatment that was administrated to the participants of the study. The treatment may be a confounding factor underestimated in the analysis. Authors should report if the intervention was the same across all participants. Please report the intensity, frequency and who administrated the behavioral treatment. If it was based on clinical practice, report what was performed. Add a paragraph in the Materials and Methods to describe the behavioral treatment

Explain the rationale for dividing the Age class in 18-24 months, 25-36 months and 37-48 months.

Discussion

Please revise the discussion removing the analytic results already presented in the Result section.

Discuss the findings in light of the behavioral treatment performed. This is of uttermost importance for interpreting the stability of the measures.

 REFERENCES

The numbers of references are not reported. Please adjust according to the journal style.  

Author Response

09-Dec-2020

Dear Editor and Dear Reviewers,

I would like to thank the editor and reviewers for your valued comments and suggestions to the article. As you requested, we made all the necessary changes in our manuscript to address the reviewers’ concerns and we detailed below how the points raised by the referees have been accommodated. We highlighted the changes within the document by using the track changes mode in MS Word. From the changes made in the revised manuscript and responses provided below, I hope you are convinced that we have adequately addressed the reviewer’s concerns and made the paper better. If there are any further questions, please feel free to let me know.

Reviewer 2

Dear Authors,

Thank you for the opportunity to revise your manuscript.

The study is interesting and well written. Authors should follow the STROBE in order to organize their manuscript.  Vandenbroucke JP, von Elm E, Altman DG, Gøtzsche PC, Mulrow CD, Pocock SJ, Poole C, Schlesselman JJ, Egger M; STROBE initiative. Strengthening the Reporting of Observational Studies in Epidemiology (STROBE): explanation and elaboration. Ann Intern Med. 2007 Oct 16;147(8):W163-94. doi: 10.7326/0003-4819-147-8-200710160-00010-w1. PMID: 17938389. There is a mixture of methods and results, as well as in the discussion section there is a repetition of the results.

Please find below my comments:

Introduction

1)When introducing the clinical description of Autism, the Authors may find interesting and cite the following studies addressing gait behaviour and postural capacity in Autism Spectrum Disorder:  

Valagussa G, Balatti V, Trentin L, Piscitelli D, Yamagata M, Grossi E. Relationship between tip-toe behavior and soleus - gastrocnemius muscle lengths in individuals with autism spectrum disorders. J Orthop. 2020 Aug 20;21:444-448. doi: 10.1016/j.jor.2020.08.013. PMID: 32982098; PMCID: PMC7493131.

Perin C, Valagussa G, Mazzucchelli M, Gariboldi V, Cerri CG, Meroni R, Grossi E, Cornaggia CM, Menant J, Piscitelli D. Physiological Profile Assessment of Posture in Children and Adolescents with Autism Spectrum Disorder and Typically Developing Peers. Brain Sci. 2020 Sep 27;10(10):681. doi: 10.3390/brainsci10100681. PMID: 32992546; PMCID: PMC7601261.

Answer: thanks for your suggestion, we cited them in line 64.

2)Lines 84-92: Please state clearly the Objectives and the Hypotheses. The paragraph is hard to follow. Moreover, in the Materials and Methods, Statistical Approach, refer the analyses to the Objectives. This will increase the readability of the manuscript.

Answer: We tried to improve the readability and understandability of this section

3)Line 95: “two-survey design” is misleading. This is a prospective observational cohort study

Answer: thanks for your suggestion; we modified in this way: “We implemented a longitudinal prospective observational cohort study”.

4) Line 112: How genetic conditions were ruled out?

Answer: each patient underwent karyotype, Xfra analysis; in favor of the reader, we added an example in the text.

5) Line 188: How sensibility, specificity, positive and negative predictive values were evaluated? Was used a ROC?  In the Results section should not be described how the analysis was performed. All the test e.g., Wilcoxon Signed Ranks Test, chi-square test should be presented here. 

Answer: done, we enlarged the analysis explanation in Statistical Approach paragraph (line 203).

6) Authors should report the behavioral treatment that was administrated to the participants of the study. The treatment may be a confounding factor underestimated in the analysis. Authors should report if the intervention was the same across all participants. Please report the intensity, frequency and who administrated the behavioral treatment. If it was based on clinical practice, report what was performed. Add a paragraph in the Materials and Methods to describe the behavioral treatment

Answer: the study aims to be an analysis of the development trajectories in a population of children mainly defined by the age variable. The therapies performed by the patients are usual treatments dictated by a medical indication in consideration to the services offered by the area; the treatments performed by the children were therapies provided by the national health system, with an average of 4 (± 1) hours per week, mainly on psychomotor skills and speech therapies. We do not feel like adding a paragraph dedicated to the therapies carried out by the patients, because the study does not want to be a comparison of effectiveness between them; however, we believe that it is right to point out the need for a better definition of the rehabilitation treatments received by these children. We therefore enlarged the explanation of the type of treatments and stated that it was homogeneous for all the enrolled children (line 161); furthermore, we included your suggestion as limitation of the study (line 435).

7) Explain the rationale for dividing the Age class in 18-24 months, 25-36 months and 37-48 months.

Answer: done, thanks for your suggestion;

Discussion

8) Please revise the discussion removing the analytic results already presented in the Result section.

Answer: done

9) Discuss the findings in light of the behavioral treatment performed. This is of uttermost importance for interpreting the stability of the measures.

Answer: see the previous answer, the study aims to be an analysis of the development trajectories in a population of children mainly defined by the age variable; the therapies performed by the patients are usual treatments, dictated by a medical indication in consideration to the services offered by the area.

 REFERENCES

10) The numbers of references are not reported. Please adjust according to the journal style.  

Answer: thanks for the suggestion. Now the reference list is correctly numbered.

Round 2

Reviewer 2 Report

Dear Authors, 

Thank you for revising the manuscript.

I have no further comments.Please carefully review the text for typos and grammatical errors.